# Nanoscale feedback control of six degrees of freedom of a near-sphere

Mitsuyoshi Kamba[1], Ryoga Shimizu[1] & Kiyotaka Aikawa [1] ✉

Manipulating the rotational as well as the translational degrees of freedom of rigid bodies has been a crucial ingredient in diverse areas, from optically controlled micro-robots, navigation, and precision measurements at macroscale to artificial and biological Brownian motors at nanoscale. Here, we demonstrate feedback cooling of all the angular motions of a near-spherical neutral nanoparticle with all the translational motions feedback-cooled to near the ground state. The occupation numbers of the three translational motions are $6 \pm 1$, $6 \pm 1$, and $0.69 \pm 0.18$. A tight, anisotropic optical confinement allows us to clearly observe three angular oscillations and to identify the ratio of two radii to the longest radius with a precision of 0.08 %. We develop a thermometry for three angular oscillations and realize feedback cooling of them to temperatures of lower than 0.03 K by electrically controlling the electric dipole moment of the nanoparticle.

The optical control of mechanical oscillators, typically possessing one translational degree of freedom, has been successful in exploring their motions at the quantum level, demonstrating various applications such as quantum transducers[1] and precision measurements[2]. Recently, observing and controlling the translational motion of near-spherical nanoparticles levitated in a vacuum has reached a low-temperature regime from which their quantum properties may be made evident[3–6]. By levitating nanomechanical oscillators, one can expect an extremely high quality factor suitable for their coherent manipulations[7–9]. At the macroscale, mechanical motions of objects are detected by accelerometers and gyroscopes, where the sensitivities of these sensors matter. By contrast, manipulating all external degrees of freedom of nanoscale objects, desired for applications such as sensing and quantum mechanical studies[7,8,10,11], has been a challenging task. The challenge at nanoscale is to detect all the minute motions with a precision sufficient for their manipulation. The remarkable progresses made with ground-state cooling of levitated objects have been particularly successful with nearly spherical particles and focused on one of their three translational motions[3–6], while the other two translational degrees are still far from the quantum regime and three rotational degrees remain uncontrolled. It is just very recently that controlling multiple translational motions of nanoparticles in the quantum regime via cavity cooling became possible[12], while feedback cooling of multiple degrees of freedom near the quantum regime has been elusive.

The rotational degrees of freedom of levitated nanoparticles has attracted attention just recently[13,14]. In this context, highly anisotropic nanoparticles, such as nanodumbells and nanorods, are expected to be a promising system for exploring fundamental physics[10,11,14–19]. Cooling of up to two librational motions, oscillations like a physical pendulum, of nanodumbells, with[20] and without[21] translational cooling, and of one librational motion of micron-scale spheres[22] have been reported. Very recently, cavity cooling of all the mechanical degrees of freedom of a nanodisk was also demonstrated[23].

Our work points out that nearly spherical nanoparticles, whose deviation from a sphere has been overlooked in previous studies, can also be a promising system for controlling all the mechanical degrees of freedom because of their slight deformations from a perfect sphere. With a tight optical confinement, such slight deformations are sufficient to enable us to observe their librational motions and characterize the shape, while the minimal deviation suppresses the decoherence rate of librational motions via photon scattering to significantly lower values than that of highly anisotropic nanoparticles. Furthermore, in comparison with the spectra of highly anisotropic particles[16,20,24], the observed simple, narrow spectra of librational motions facilitate controls over their multiple degrees of freedom in the quantum regime.

[1]Department of Physics, Tokyo Institute of Technology, 2-12-1 Ookayama, Meguro-ku, 152-8550 Tokyo, Japan. ✉e-mail: aikawa@phys.titech.ac.jp

## Results

### Observation of the librational and translational motions

In our experiments, we trap nearly spherical silica nanoparticles in a one-dimensional optical lattice formed in the vacuum chamber at a pressure of $6.5 \times 10^{-7}$ Pa[25,26] (Fig. 1). The nanoparticle is neutralized[6,27,28] and has an average radius of 176(3) nm. The translational motion along the optical lattice ($z$ direction) is cooled to an occupation number of $n_z = 0.69(18)$ via optical feedback cooling realized by controlling the optical gradients[6,29], as shown in the power spectral density (PSD) obtained with photodetectors (Fig. 2a). We apply a similar approach also for cooling the translational motions in the $x$ and $y$ directions, in contrast to previous studies, where motions in two unfocused directions are cooled via parametric feedback cooling (PFC)[3–6]. We introduce additional two beams providing tunable optical gradients and modulate the intensity ratio of the two beams such that they exert feedback forces in both $x$ and $y$ directions. In this manner, the translational motions along the $x$ and $y$ directions are cooled to occupation numbers of $n_x = 6(1)$ and $n_y = 6(1)$, respectively, which are more than one order of magnitude lower than obtained with PFC[25,30–32] (Fig. 2b).

In a frequency range between 10 and 70 kHz, additional narrow peaks are observed (Fig. 2c). These peaks are visible only at low pressures, where the broad spectra of the translational motions are minimized by feedback cooling. The frequencies are proportional to the square-root of the laser power and vary among trapped particles. We identify that these peaks originate from the three librational motions of the trapped nanoparticles and their higher order signals. These three peaks are correlated with each other in terms of both the amplitude and the frequency (see Supplementary information), suggesting rich dynamics such as gyroscopic and precession motions[20,33–36]. Theoretically, it was suggested that librational motions are independent only in the deep-trapping regime where their amplitudes are sufficiently small[36].

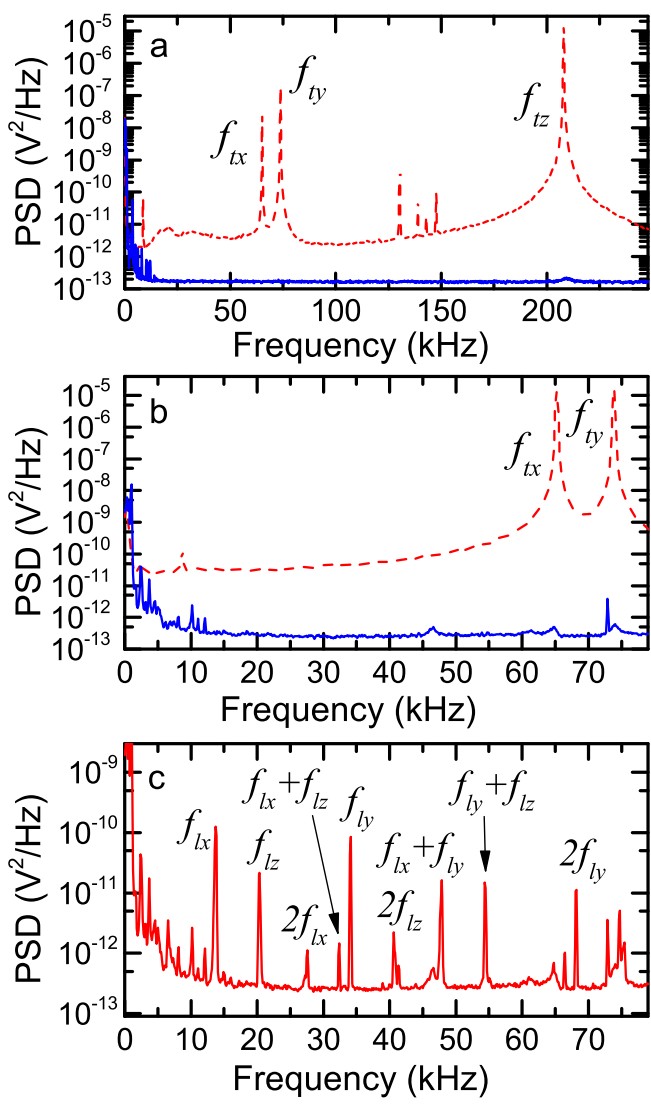

**Fig. 1 | Experimental system. a** An electron microscope image of the sample of silica nanoparticles used the present study. **b** An overview of our experimental setup. A near-spherical nanoparticle is trapped in an optical lattice. The translational motions along the $x, y, z$ axes have frequencies of $f_{tx}, f_{ty}, f_{tz}$, respectively, and are cooled via optical feedback cooling. The librational motions around the $x, y, z$ axes have $f_{lx}, f_{ly}, f_{lz}$, respectively, and are electrically controlled. Three photodetectors (PDs) are used for observing and controlling six degrees of freedom. The trapping laser is polarized along the $x$-axis. Two angles $\theta, \phi$ defines the orientation of the electric dipole moment of the trapped particle, indicated by the blue arrow. **c** Comparison of two configurations for trapping an anisotropic nanoparticle in an optical lattice. The mechanical potential energy is lowered when the long axis of the nanoparticle is perpendicular to the direction of the optical lattice because then the interaction of the nanoparticle with the light is stronger.

**Fig. 2 | PSDs of the motions of the trapped nanoparticle. a** PSD of the PD1 signal. The translational motion at $f_{tz}$ is cooled to a temperature of 12(2) $\mu$K. **b** PSD of the PD3 signal. The translational motions at $f_{tx}, f_{tx}$ are cooled to temperatures of 19(3) $\mu$K, 24(4) $\mu$K, respectively. In addition, all the peaks arising from librational motions are extinguished. For both panels, the blue solid and red dashed lines show the PSDs with and without six-dimensional feedback cooling, respectively. The uncooled curve is obtained at 5 Pa. The peak at 73 kHz is an intrinsic laser noise. **c** PSD of the PD3 signal with feedback cooling only for the translational motions, where three librational motions and their higher order signals are remaining. The assignment on the higher order signals are also indicated.

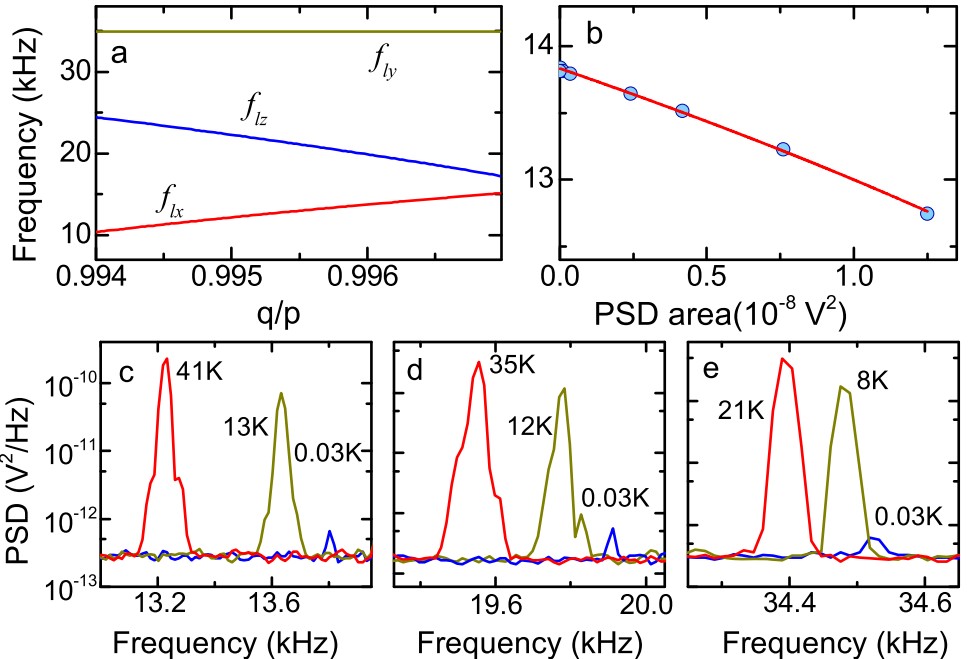

**Fig. 3 | Variations of librational frequencies. a** Calculated librational frequencies with respect to the radius along the $y$-axis $q/p$. The radius $r$ is set to 0.9914$p$. The librational frequencies are sensitive to the geometry of the trapped nanoparticle. **b** Measured frequency variation of $f_{lx}$ as a function of the PSD area. The solid line is a fit with Eq.(1). The librational frequency varies with the temperature of the librational motion, which is proportional to the area of the PSD. See Supplementary Fig. 3 for $f_{ly}$ and $f_{lz}$. For this measurement, other librational motions are feedback-cooled in advance to minimize the correlation effect. **c** PSD near $f_{lx}$ for three temperature values. **d** PSD near $f_{lz}$ for three temperature values. **e** PSD near $f_{ly}$ for three temperature values. The lowest temperatures are lower than 0.03 K for any direction.

## Precise determination of the shape of a near-sphere

The observed librational motions arise from the orientational confinements in three rotational angles around the $x, y, z$ axes. In our study, there are two independent mechanisms for yielding such confinements. The first mechanism is a tendency that the longest axis of the particle is aligned with the polarization of the light[37,38], which has been experimentally observed with highly anisotropic particles such as nanodumbells[15,20,21,37]. The second mechanism is an effect originating from the anisotropy of the optical trap. Under the generalized Rayleigh-Gans approximation, where we consider the inhomogeneous electric field of the trapping laser inside a nanoparticle and assume that the light scattering by the nanoparticle does not modify the local electric field, the mechanical potential energy of a non-spherical nanoparticle in an anisotropic trap $U$ is obtained by integrating the local interaction potential energy density over the volume of the nanoparticle[10,35,36] (see Methods for more details). $U$ is dependent on the relative angle between the nanoparticle orientation and the optical trap and is minimized when the longer axis of the particle is aligned to the orientation with the lower translational oscillation frequency (Fig. 1). In other words, the nanoparticle is aligned such that it experiences a higher light intensity. Theoretical studies on the librational motions of various geometries under this approximation have been reported[10,11,34–36,39]. To our knowledge, librational motions with the latter mechanism have not been experimentally observed.

Considering the two mechanisms, under an assumption that the trapped nanoparticle is an ellipsoid with radii of $p, q$ and $r$ as defined in Fig. 1b, we obtain the expressions of the librational frequencies $f_{li0}$ around the $i$-axis when the oscillation amplitudes are small, where $i \in \{x, y, z\}$, as functions of $p, q$ and $r$ (see Methods). Calculated librational frequencies with respect to $q/p$ show that they are sensitive to the anisotropy of the trapped nanoparticles (Fig. 3a; see Supplementary Fig. 2 for the plot with respect to $r/p$). Due to the strong confinement with the light polarization, the largest axis of the particle aligns with the light polarization. In addition, the strong trap

anisotropy in the $yz$ plane, that is, $f_{ty} < f_{tz}$, suggests that the second elongated axis aligns with the $y$-axis. Thus, we expect that a configuration of $p > q > r$ provides the minimum potential energy.

Experimentally, we observe three frequencies of 13.9 kHz, 19.9 kHz, and 34.5 kHz when their amplitudes are sufficiently small. By minimizing the deviation between observed and calculated frequencies, we determine two radii to be $q = 0.9961(8)p$ and $r = 0.9914(8)p$, with which we can reproduce observed frequencies within 0.7%. The precision in determining $q/p, r/p$ is limited mainly by systematic uncertainties in the radius and the refractive index and is about 800 ppm (see Methods). The precision can be comparable to the size of atoms because our measurement is based on the averaged interaction of light and many atoms. The demonstrated precision suggests a novel approach to precisely characterize the three dimensional shape of trapped nanoparticles without relying on electron microscopes.

## Thermometry of the librational motions

We find that librational frequencies are also sensitive to the temperatures of the librational motions (Fig. 3c–e). This is because the depths of the potential energies for librational motions are of the order of $k_B \times 10^2$ K, where $k_B$ is the Boltzmann constant, and the librational motions can be readily excited to amplitudes that can experience the nonlinearity of the potential. Such a situation is in contrast to previous studies on the nonlinearity observed with the translational motions of nanoparticles[40] and with the librational motions of nanodumbells[20], where the potential depths are more than $k_B \times 10^4$ K. Under an assumption that the time variation of the amplitudes of three librational motions are negligible, the average oscillation frequency with a finite amplitude for the $i$ direction can be written as

$$f_{li} = f_{li0}\sqrt{\frac{\sin\sqrt{2}\beta}{\sqrt{2}\beta}} \qquad (1)$$

with $\beta = \sqrt{2 - \sqrt{4 - 2k_B T_{li}/(\pi^2 I_i f_{li0}^2)}}$, where $T_{li}$ and $I_i$ denote the temperature of the librational motion and the moment of inertia around the $i$-axis. As shown in Fig. 3b for $f_{lx}$, the derived expression (1) is in good agreement with experimentally observed frequency variations with respect to the area of the PSD, which is proportional to the motional temperature[21]. For the determination of the radii $q, r$, we use $f_{li0}$ obtained by the fits.

An important suggestion here is that such a measurement can provide a direct, independent thermometry of the librational motions, i.e. to obtain a conversion between a signal voltage and the temperature. This is because the extent of the nonlinearity directly reflects the absolute magnitude of the angular deviation. A similar idea has been employed for calibrating the translational motions of levitated particles[41,42]. The temperature values obtained in this approach are shown in Fig. 3c, d, e. Note that we observe considerable nonlinear frequency shifts at large libration amplitudes with temperatures of higher than 1 K. In many previous works with nanoparticles, thermometry has relied on the thermalization at high pressures to the temperature of background gases[7,43]. However, establishing an independent method of thermometry is crucially important because thermalization measurements are always accompanied by large thermal fluctuations.

### Feedback cooling of the librational motions

We manipulate the orientation of the nanoparticle by applying electric fields on it. Even though the trapped nanoparticle is neutralized, they can have a charge distribution over its surface and/or inside its volume, yielding a finite dipole moment[44]. This fact implies that we can exert a feedback torque proportional to the angular velocity on nanoparticles by applying an electric field synchronized to the librational motion[22]. Such a cooling scheme is called cold damping and has been shown to be more efficient than parametric feedback cooling[25,45,46]. We observe that cooling is realized only when the phase of the applied electric field is chosen appropriately. When feedback electric fields include three independent signals synchronized to three librational frequencies, we are able to completely extinguish all the peaks from librational motions as well as their higher order signals (Fig. 2). Due to the correlation among the librational motions, we observe that extinguishing two librational motions decreases the remaining motion as well, but does not extinguish it perfectly. Such a behavior implies that the three librational motions become independent in the deep-trapping regime when their amplitudes are small[36]. The lowest temperatures are estimated to be lower than 0.03 K for each librational motion, limited only by the noise floor for observing the motions. The fundamental limit of cooling is determined by the compromise between the intrinsic heating rate, which will be discussed later, and the noise introduced by feedback[25,45,46]. The obtained temperatures are comparable to or lower than those obtained with nanodumbells[20,21], which are realized with PFC.

To gain further insights on the dipole moment of nanoparticles, we explore the cooling dynamics. From the equation of the librational motion, we obtain an expression for describing the time evolution of $T_{li}$ in the presence of a feedback torque:

$$T_{li}(t) = \left( \sqrt{T_{li0}} - C_i(t - t_0)/2 \right)^2 \qquad (2)$$

where $C_i = dE_0\eta_i/\sqrt{2k_B I_i}$ is a damping rate due to feedback cooling with $d$, $E_0$, and $\eta_i$ being the dipole moment, the electric field amplitude, and the angle factor considering the angle between the dipole moment and the electric field (see Methods). 

A typical time evolution of $T_{lx}$, when the feedback signal is applied, is show in in Fig. 4a. We observe that the time variation under feedback cooling is in good agreement with the theory, suggesting that feedback cooling works as expected. We confirm that the damping rate due to feedback cooling is proportional to the applied electric

field amplitude (Fig. 4b). From the three values of the slopes $C_i/E_0$ (see Methods), we deduce that the trapped nanoparticle has a dipole moment with a magnitude of $d = 2p \times 1.92(13)e$, with $e$ being the elementary charge, and an orientation defined by $\theta = 31(1)°$ and $\phi = 65(2)°$. The dipole moment is nearly constant for half a day. Given that the initial charge distribution before neutralization is of the order of $10e^6$, the obtained magnitude is consistent with an interpretation that the dipole moment originates from a few elementary charges remaining over the surface even after neutralization. Exploring the stability of the dipole moment over a longer period may help understanding the origin of the dipole moment[22].

### Heating dynamics of the librational motions

We also explore the heating dynamics of the librational motions by observing the time variation of the amplitudes of the librational motions after feedback cooling is turned off (Fig. 4c for $f_{lx}$). After averaging over many experimental runs, we observe a linear increase in the temperature. The measured heating rates are 2.6(2) mK/s, 2.1(1) mK/s, and 2.3(1) mK/s for $f_{lx}, f_{ly}$, and $f_{lz}$, respectively. The low heating rate is also reflected in the narrow spectral width of the PSD (Fig. 2c), where the observed width of around 10 Hz is not limited by the heating rate but rather reflects the Fourier limit and fluctuations in the laser intensity and in the amplitude of librational motions. These measured heating rates are two orders of magnitude lower than previously measured values for the librational motions of nanodumbells[21] and typical heating rates for the translational motions of optically trapped nanoparticles[26,45], both of which are limited by photon scattering at high vacuum.

The observed slow heating dynamics reflects the fact that the photon recoil torque strongly depends on the geometry of the particle and is equal to zero for spherical particles[47]. We compare the measured heating rates with calculated values obtained as the sum of photon recoil heating and background gas heating (see Methods), as shown in Fig. 4d, and find a good agreement. The agreement shows that heating is dominated by background gas collisions at the current pressure. The agreement between experiments and calculations also confirms the validity of the thermometry based on the nonlinearity of the trap.

Given that slow decoherence is a crucial ingredient for quantum mechanical experiments, identifying a highly coherent system is an important task. As shown in Fig. 4d, by decreasing the pressure by two orders of magnitude, we expect to reach a regime where the decoherence of the librational motion is only limited by very slow photon recoil heating. At such a regime, the number of coherent librational oscillations, during which the phonon occupation number is preserved, is expected to be more than 2000 for $f_{ly}$, which is more than two orders of magnitude larger than the value expected with the translational motions[3].

## Discussion

The present study is important in the following aspects. First, even though the nearly spherical geometry does not seem optimal for observing and controlling the librational motions, we show that all the librational motions can be clearly observed and cooled to temperatures of below 0.03 K. Second, we establish methods to characterize trapped nanoparticles precisely in terms of the geometry, the dipole moment, and the temperatures of librational motions. Third, because of the nearly spherical shape, heating of librational motions via photon recoils is negligible, and is only limited by very slow heating via background gases. Fourth, the higher order signals of librational motions often interfere with frequencies of the translational motions, thereby prohibiting efficient feedback cooling of the translational motions. We demonstrate that all the signals arising from the librational motions can be extinguished and are not an obstacle to cooling all the translational motions to near the ground state.

Characterizing the geometry of nanoscale objects has been a crucial issue in a wide variety of applications in biology, chemistry,

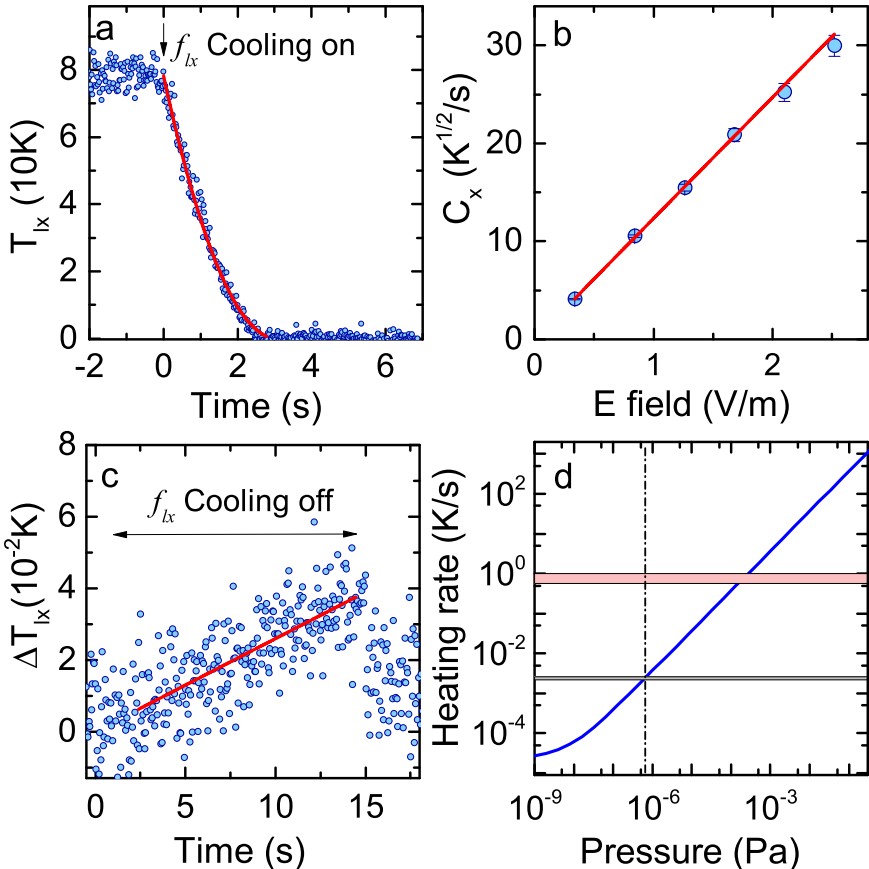

**Fig. 4 | Dynamics of librational motions. a** Time evolution of $T_{lx}$ with electric feedback cooling. The solid line is a fit with eq.(2). **b** The damping rate with respect to the applied electric field. The error bar is a statistical error in determining the damping rate. The solid line is a linear fit. Plots for $T_{ly}$ and $T_{lz}$ are provided in Supplementary Fig. 6. **c** Time variation of $T_{lx}$ after electric feedback cooling is turned off. The trace is averaged over 64 experimental runs. We observe slow heating due to background gas collisions. The solid line is a linear fit. Plots for $T_{ly}$ and $T_{lz}$ are provided in Supplementary Fig. 7. **d** Calculated heating rate of librational motions as a function of the pressure. Heating via background gas collisions is dominant above $10^{-8}$ Pa. The range of the observed heating rates is indicated by the gray area, while heating rates previously found for translational motions[26,45] and librational motions[21] are indicated by the red area. The dot-dashed line shows the pressure at which the present work is performed.

physics, and engineering[48]. Electron microscope imaging has been extensively employed[49]. Optical measurements of the shape of trapped nanoparticles has been reported[16,24,50]. Our approach is particularly suited for near-spherical particles and provides a novel route to optically measure their shape with a precision of 0.08 %. Such a precision indicates discerning the difference in diameter of 0.3 nm, which is smaller than a recent demonstration in a related setup[24] by one order of magnitude and is close to the precision of 0.12 nm obtained with latest electron microscopes[51]. Our approach may be extended to particles with a geometry far from a sphere if an appropriate model for describing their motions in an optical trap is developed.

Our study is an important building block towards quantum mechanical experiments with levitated nanoparticles. There has been various proposals to observe quantum superposition states of the motions of nanoparticles, including approaches based on an optical diffraction grating[52] and quantum state tomography via time-of-flight measurements[53]. However, given that the particle is not a perfect sphere, motions in other degrees of freedom, in particular librational motions, can readily obscure the minute effect of the motion cooled to the ground state. In this perspective, freezing all the degrees of freedom will be an essential ingredient in future studies[7,8,10,11].

The quantum mechanical behaviors associated with the librational motions are also an intriguing subject. It has been proposed that orientational quantum revivals can be observed with free-falling nanorods[18]. The observed very slow heating rate of our system, combined with the demonstrated low temperatures of librational motions,

which are lower than assumed in the proposal[18], suggests that nearly spherical nanoparticles can also be a promising candidate for investigating quantum physics with the orientation of nanoparticles. Although the temperatures of librational motions achieved in the present study are limited by the signal-to-noise ratio (SNR) of our experimental setup, enhancing the observation sensitivity via dedicated experimental improvements will enable us to further approach the quantum regime[54]. The ability of feedback control on the rotational degrees of freedom also opens the door to studies on information thermodynamics with nanomechanical heat engines both at the classical and at the quantum regime[55].

Note added: After the submission of the present study, related works on cavity cooling of six degrees of freedom of a nanodisk[23] and on cavity cooling of the center-of-mass motions to the ground state in two dimensions[12] are published.

## Methods
### Experimental setup
A single-frequency laser at a wavelength of 1550 nm and with a power of 176 mW is focused with an objective lens (NA = 0.85) and is approximately quarter of the incident power is retro-reflected to form a standing-wave optical trap (an optical lattice). The beams for cooling the translational motions in the $x$ and $y$ directions have a power of about 1 mW in total. We load nanoparticles by blowing up silica powders placed near the trapping region with a pulsed laser at 532 nm at pressures of about 400 Pa. At around 350 Pa, we apply a positive high

voltage to induce a corona discharge and provide a positive charge on the nanoparticle. Then we evacuate the chamber with optical feedback cooling for the translational motions and neutralize the nanoparticle via an ultraviolet light at around $2 \times 10^{-5}$ Pa.

In the present study, three photodetectors are used for observing and controlling the motions of a nanoparticle as shown in Supplementary Fig. 1. For observing the translational motion in the $z$ direction, PD2 works as an in-loop (IL) detector, while PD1 works as an out-of-loop detector for properly estimating the temperature. For the $x$ and $y$ directions, PD3 works as an IL detector, which was also used for estimating the translational temperatures in $x$ and $y$ directions. Regarding feedback cooling in the $x$ and $y$ directions, we work in a regime where noise squashing is not observed[45,46]. Therefore, the temperature estimations with an IL detector is expected to be valid.

## Observation and feedback cooling of librational motions
The librational motions are observed with PD1 and PD3 (Supplementary Fig. 1). The peaks at $f_{lx}$ and $f_{ly}$ are clearly observed with PD3, while the peak at $f_{lz}$ is clearly observed with PD1. The feedback signals are obtained from oscillators phase-locked to the signals from PD1 and PD3. The relative phases between the oscillators and the PD are adjusted to achieve maximum cooling efficiencies in each direction. The magnitudes of electric fields produced with the electrodes are estimated by using a finite-element matrix simulation of electric fields via COMSOL Multiphysics.

## Estimation of the mass and the temperature of nanoparticles
We estimate the density and the radius of the trapped nanoparticle via the two independent measurements. First, we measure the heating rate at around 5 Pa, which is given by the background gas collisions[25,30,32]. We measure the pressure with an accuracy of 0.5 % via a capacitance gauge. Second, we measure the heating rate at around $1 \times 10^{-6}$ Pa, which is determined dominantly by photon recoil heating and is more sensitive to the radius than the heating rate at 5 Pa. By combining these results, we determine the radius and the density of the nanoparicle to be 176(3) nm and $2.27(4) \times 10^3$ kg/m$^3$.

The translational temperatures are obtained by comparing the areas of the PSDs with and without cooling, as has been performed in previous studies[25,30,32]. To avoid the influence of the increase in the internal temperature of nanoparticles at high vacuum due to laser absorption[56], we take the uncooled data at around 5 Pa. We find that the typical thermal fluctuation of the area of the PSD is lower than 10 % for both cooled and uncooled data. Thus, we estimate the systematic error in determining the temperatures of translational motions to be about 14 %.

We find that the thermalization method does not provide reliable temperature estimations of librational motions for the following reason: at high pressures, the librational motions are hidden in the broad spectra of the translational motions, while at low pressures the time scale of the amplitude variation is so long that we cannot identify at which voltage the amplitude settles to background gas temperatures.

## Calculations of librational frequencies
By considering the two mechanisms of the orientational confinement, i.e. the light polarization and the trap anisotropy, we obtain the expressions for the three librational frequencies as

$$f_{lx0} = \sqrt{\frac{(f_{tz}^2 - f_{ty}^2)(q^2 - r^2)}{q^2 + r^2}}$$

$$f_{ly0} = \sqrt{\frac{(f_{tz}^2 - f_{tx}^2)(p^2 - r^2)}{p^2 + r^2} + \frac{10U_t(\alpha_x - \alpha_z)}{4\pi^2 m(p^2 + r^2)\alpha_x}} \quad (3)$$

$$f_{lz0} = \sqrt{\frac{(f_{ty}^2 - f_{tx}^2)(p^2 - q^2)}{p^2 + q^2} + \frac{10U_t(\alpha_x - \alpha_y)}{4\pi^2 m(p^2 + q^2)\alpha_x}}$$

where $m$, $U_t$, and $\alpha_i$ denote the mass of the particle, the potential depth of the translational motions, the polarizability along the $i$-axis. These expressions agree with a theoretical formalism provided in ref. 35 when the radius is sufficiently smaller than the wavelength and the beam waist. The presence of higher order terms shifts the frequencies by about 4 %. For the analysis of our measurement results, we include higher order terms that are ignored in the above expressions (see Supplementary information). An alternative route to obtain the optical torque is to integrate Maxwell's stress tensor over an infinite volume[35,36]. A comparison between the two approaches is provided in ref. 35. The latter approach is more suitable for particles with a size comparable to the wavelength, while in our case $R$ is approximately 10 % of the wavelength and validates the use of the former approach.

## Frequency variation due to nonlinearlity
The detailed derivation of Eq.(1) is provided in supplementary information. Here we briefly discuss the derivation. Given that librational motions are observed with narrow spectral widths and the time variation of their amplitudes are very slow, we can safely assume the tilt angle follows $\psi_i(t) = A(t)\sin(2\pi f_{li}t)$ with $dA/dt \ll 2\pi A f_{li}$. We then obtain the average oscillation frequencies with a finite amplitude $A$ as

$$f_{li} = f_{li0}\sqrt{\frac{\sin\sqrt{2}A}{\sqrt{2}A}} \quad (4)$$

The temperatures of librational motions are given by the sum of the potential energy and the kinetic energy:

$$k_B T_{li} = \frac{1}{2}I_i(2\pi f_{li0})^2\left(\frac{A}{2\sqrt{2}}\sin\sqrt{2}A + \sin^2\frac{A}{\sqrt{2}}\right) \quad (5)$$

which can be used to relate $A$ and $T_{li}$ and we obtain Eq.(1).

## Determination of radii
To determine the radii which minimize the difference between experimentally observed librational frequencies $f_{li0}^{obs}$ and calculated frequencies $f_{li0}^{cal}$, we define a function indicating the extent of the deviation as follows:

$$\delta = \sum_i \left(\frac{f_{li0}^{cal}}{f_{li0}^{obs}} - 1\right)^2 \quad (6)$$

Supplementary Fig. 4 shows calculated values of $1/\delta$ as functions of $q/p$ and $r/p$ near the minimum of $\delta$. We find that $\delta$ takes a minimum value with a specific set of $(q, r)$ values. By integrating $1/\delta$ in each dimension, we obtain Lorentzian-like profiles as shown in Supplementary Fig. 5. We fit Lorentzian functions on these profiles and determine the values of $(q, r)$. The precision in determining the peaks is limited by the fitting procedure to about 10 ppm. Because of the asymmetric profile, the center position obtained from the fit weakly depends on the range of the fit. Developing an appropriate two-dimensional fit may improve the precision, which will be an interesting future study.

Apart from the uncertainty stated above, we identify several sources of systematic uncertainties listed in Supplementary Table 1. The dominant contributions are the uncertainties in the refractive index and in the measured radius. The refractive index of the sample used in the present study is measured to be 1.43025(32) by observing the transmissivity of the mixtures of the sample and refractive index liquids, while we derive a systematic uncertainty of 0.76 % from the measurement (see supplementary information for more details).

## Time evolution of the amplitude of librational motions under feedback cooling

The detailed derivation of Eq.(2) is provided in supplementary information. Here we briefly discuss the derivation. Under an assumption that the time variation of the amplitudes of three librational motions are negligible, the equation of librational motions around the $i$-axis is given by

$$I_i \frac{\mathrm{d}^2 \psi_i}{\mathrm{d}t^2} + I_i \gamma_i \frac{\mathrm{d}\psi_i}{\mathrm{d}t} + \frac{1}{2} I_i (2\pi f_{li0})^2 \sin 2\psi_i = 0 \qquad (7)$$

where $\psi_i$ and $\gamma_i = \mathrm{d}E_0 \eta_i / (2\pi I_i A f_{li0})$ are the libration angle and the damping rate around the $i$-axis, respectively. Because of the assumption that the time variation of their amplitudes are very slow, implying $\psi_i(t) = A(t)\sin(2\pi f_{li}t)$ with $\mathrm{d}A/\mathrm{d}t \ll 2\pi A f_{li}$, we arrive at the equation for $T_{li}$:

$$\frac{\mathrm{d}T_{li}}{\mathrm{d}t} = -C_i \sqrt{T_{li}} \qquad (8)$$

whose solution is given by Eq.(2). Here we assume that the influence of the nonlinearity of the angular potential on the dynamics is negligible, which is a good approximation in our measurements. Eq.(2) is valid at $t \le t_0 + 2\sqrt{T_{li0}}/C_i$ because the feedback signal is not locked to the position signal once $T_{li}$ approaches to zero. Note that Eq.(2) differs from an exponential decay observed in previous studies because the feedback signal is obtained from an oscillator with a constant amplitude, instead of utilizing a filtered photodetector signal[6].

## Measurements of the damping rate and the heating rate

The amplitude of the signal of the librational motion is extracted with an lock-in amplifier (MFLI, Zurich Instruments). Although the signal obtained in this manner is essentially the same as integrating the PSD, observing the time evolution of the signal amplitude is easier than processing the PSD. To derive the dipole moments and two angles $\theta, \phi$ to define its orientation, we use the angle factors in our system represented by $\theta, \phi$ as $\eta_x = \sin\theta\sin\phi$, $\eta_y = \cos\theta\cos(\pi/4)$, and $\eta_z = \sin\theta\sin(\phi - \pi/4)$. The slopes obtained in measurements in Fig. 4b and in Supplementary Fig. 6 are $C_x/E_0 = 12.4(2)$, $C_y/E_0 = 16.1(2)$, and $C_z/E_0 = 4.7(3)$ all in unit of $\mathrm{m}\sqrt{\mathrm{K}}/(\mathrm{Vs})$.

We note that the translational motions in the $x$ and $y$ directions are cooled via parametric feedback cooling during the measurements on the heating rate. With optical cold damping on the $x$ and $y$ directions, we observe heating rates of the order of 1 K/s, presumably because the two cooling beams are slightly misaligned and the modulation of their relative intensity provides a net torque on the nanoparticle. Because such heating rates are much lower than the damping rates achieved with feedback, feedback cooling of librational motions successfully extinguish all the peaks as shown in Fig. 2. For future applications based on coherent librational oscillations, minimizing such a heating effect will be important.

## Calculations of the heating rate

We consider two heating mechanisms, photon recoil heating and background gas collisions. To our knowledge, general expressions including both mechanisms for an ellipsoid with $p \ne q \ne r$ have not been reported. We calculate photon recoil heating using the expression in ref. 47 for an oblate particle with an assumption of $p = q$ and $r = 0.9914$. We estimate the heating rate via background gases via an expression for a sphere in Ref. 57. Background gas heating depends on the temperature of surrounding gases, which can be higher than room temperature because of the elevated internal temperature of trapped nanoparticles via laser absorption[56]. We estimate the temperature of surrounding gases to be about 340 K (see supplementary information for more details).

## Data availability

All data that support the findings of the study are provided in the main text and in the Supplementary Information. Raw data are available from the corresponding authors K. A. upon request.

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

## Acknowledgements

The authors thank M. Kozuma and T. Mukaiyama for fruitful discussions. We are grateful to T. Tsuda for his experimental assistance. M.K. is supported by the establishment of university fellowships towards the creation of science technology innovation (Grant No. JPMJFS2112). This work is supported by the Murata Science Foundation, the Mitsubishi Foundation, the Challenging Research Award, the 'Planting Seeds for Research' program, Yoshinori Ohsumi Fund for Fundamental Research, and STAR Grant funded by the Tokyo Tech Fund, Research Foundation for Opto-Science and Technology, JSPS KAKENHI (Grants No. JP16K13857, JP16H06016 and JP19H01822), JST PRESTO (Grant No. JPMJPR1661), JST ERATO (Grant No. JPMJER2302), JST COI-NEXT (Grant No. JPMJPF2015) and JST CREST (Grant No. JPMJCR23I1).

## Author contributions

K.A. and M.K. designed and built the experiments. M.K. performed measurements and analyzed the data. R.S. performed numerical simulations. All authors discussed the results and contributed to writing the manuscript.

## Competing interests

The authors declare no competing interests.
