## [Peer Review File · Nature Communications]

REVIEWER COMMENTS

Reviewer #1 (Remarks to the Author):

The manuscript by Kamba, Shimizu, and Aikawa discusses the optical levitation and feedback-cooling of a slightly non-spherical silica nanoparticle in an optical lattice. The authors succeed in cooling its three centre-of-mass modes close to the quantum ground state (with one mode reaching an occupation less than unity), and they argue that the three librational modes reach temperatures below 100 mK.

The experimental techniques and methodology are clearly presented and understandable also to non-experts. The presented techniques to control and cool all six mechanical degrees of freedom of a nearly spherical object are highly interesting for researchers in the field of levitated nanoparticles and might well be an important step towards quantum experiments with such objects. My main concern is that the theoretical analysis, especially of the rotational motion, is unclear and questionable. Since this analysis lies at the basis of how the data is interpreted, I cannot recommend publication of the manuscript at this point, but ask the authors to clarify the following points:

1. One of the main claims of the manuscript is that all three angles of a nearly spherical particle can be deeply trapped in a linearly polarized optical field if the tweezer is sufficiently anisotropic. While I can believe that the authors indeed observe trapping of all three rotational degrees of freedom, I am not convinced by this mechanism. I thus believe that this point requires further strengthening.

a. Have the authors performed an independent check to verify that the three observed peaks are indeed due to rotations of the particle? Such a check could for instance use that the three angles are not independent [as Eq. (7) suggests; see below] but coupled non-linearly due to the gyroscope effect. Such gyroscopic coupling is one of the main signatures of rotations and is limiting other rotational cooling experiments thus far [PRL 127, 123605 (2021); PRR 2, 043054 (2020); Nat. Phys. <https://doi.org/10.1038/s41567-023-02006-6> (2023)].

b. If the authors' interpretation is correct, the trapping frequencies should vary with the root of the tweezer power (as is also the case for the centre-of-mass frequencies). Has this been verified experimentally? Admittedly, this check would not prove the proposed mechanism, but it would at least rule out that the observed alignment is due to, e.g., an offset electrostatic field coupling to the permanent dipole moment of the particle.

c. The authors' interpretation that the particle aligns with the trap anisotropy is based on Eq. (SI_4) in the SI. What is the theoretical foundation of this equation? To derive it, the authors integrate the

centre-of-mass trapping potential over the particle volume, arguing that this yields the ‘trap potential energy at which the particle is tilted’. This statement makes no sense since in this equation x , y and z are the centre-of-mass coordinates of the particle, rather than field coordinates. If the authors’ argument is that the electric field cannot be approximated as homogeneous over the extension of the particle (i.e., that the Rayleigh approximation fails), then this should be accounted for by a proper generalization of the optical torques derived for instance in [PRA 103, 043514 (2021)].

2. Throughout the manuscript, the authors treat the three angles as independent degrees of freedom. This is incorrect in general due to the gyroscope effect and only justified for deeply trapped rotations (librations). I am wondering why neglecting this coupling is justified even when non-linearities of the trapping potential become relevant. Specifically, if the equipartition theorem holds, as assumed in the analysis, this coupling should be on the same order of magnitude as the non-linearities of the trapping potential.

3. The discussion of the rotor dynamics in presence of trapping non-linearities is confusing because it does not distinguish clearly between fundamental parameters and time-averaged, effective quantities in the non-linear regime. For instance, I would propose to refer to Eq. (1) as the average frequency (or similar) rather than as the ‘librational frequency’ (which is already defined as ω_0). Additional (minor) points:

a. In the second equation in the SI I would expect a factor of 2 in front of the amplitude A in the numerator of the fraction because of the $\sin(2\psi)$ term on the right-hand side of the first equation in the SI.

b. How is the replacement of the time-dependent sine function in the SI = $1/\sqrt{2}$ justified? This seems very arbitrary, given that averaging the sine function over one oscillation period yields zero.

4. The potential energy of a non-spherical particle in a laser field does not decompose into a sum as assumed in Eqs. (SI_1) – (SI_3), unless the angles can be harmonically expanded. For the correct expression (in the Rayleigh regime), see for instance [PRA 103, 043514 (2021)].

5. The authors might want to consider referencing the works from Peter Barker’s group on 6D cooling of nanoparticles [Nat. Phys. <https://doi.org/10.1038/s41567-023-02006-6> (2023)] and on measuring the shape of optically levitated nanorotors [APL 121, 221102 (2022)].

Reviewer #2 (Remarks to the Author):

Overall this is an interesting demonstration of control not only of the translational, but also the rotational, degrees of freedom of an optically trapped nanoparticle in vacuum. While previous work has largely focused on the translational degrees of freedom, this study extends this to full control (and cooling) of the angular orientations as well. Control of angular degrees of freedom is likely to be of high importance for future work in the field, and as the authors point out these DOF may have certain advantages for control and measurement of massive objects in the quantum regime.

While I believe the overall results are of interest and the methods sound, I recommend the authors consider the following comments/questions:

Line 50: An error on the average radius of only 4 nm is derived, but as discussed in the supplementary material this number appears to assume a density of 2.2 g/cm^3 with negligible error. The authors should comment on how this density is known with sufficiently small error, in particular since previous work has typically found lower density for the spherical nanoparticles used in most experiments, and exact measurement of that density is challenging. If there is an error on density, how would it translate to the predicted frequencies used to constrain the particle sphericity?

Line 86 and elsewhere: The paper claims to measure the fractional ellipticity (q/p , r/p) at the 10 ppm level by comparing the observed frequencies and a model. However, I don't see any discussion of systematic errors relevant for this measurement. Instead the claim is it is primarily limited by the accuracy of fitting the frequency in the PSD. Could the authors comment whether e.g. small deviations in the polarization, shape of the optical potential, etc could provide relevant systematic errors for these fits at the ppm level? Have these systematics (or other dominant ones) been constrained and ruled out?

Line 134-138: The authors provide a fairly precise measurement of the dipole moment and its orientation. However, previous studies have found significant variations in the apparent dipole moment over time in larger optically trapped microparticles (e.g. Ref. 29). Since some conductivity is likely possible for charge within the particle, at least on long time scales, information quantifying the stability of such charge distributions may be useful if available. Such time variation may also be relevant to whether charges are primarily located on the surface as suggested in Lines 136-138.

Reviewer #3 (Remarks to the Author):

See attached file.

This manuscript presents work on the observation of the angular motion of a nearly-spherical nanoparticle. It involves cooling, heating measurements and thermometry. On a first read I was extremely impressed, it is nice work building on a really nice body of previous results. However, on a more detailed reading I developed many concerns.

As a very general point, I think the authors need to consider what the point of this paper is. They do not achieve temperatures lower than others (for their librations), so this is not the selling point; it's very far from the ground state of the librations. The cooling mechanism seems novel, but it's barely explained! The size estimation I think has major issues, similarly the thermometry is highly specific and has issues. So, is the selling point that they are working with spheres, but they have achieved enough control and detection efficiency to realise that there's no such thing as a sphere? If so, why is this interesting (I'm not sure I buy the argument it's "better" for anything, but it's certainly an observation worth making)?

The figures in this manuscript are of a generally low quality. All figures are too small, with crowded axes labels. Figure 1 is ok. Figure 2 is poor, very crowded, and barely shows the librational frequencies (just as an inset with many other unexplained peaks) even though this is the core of the paper. I feel that the figures do not convey the core messages of the manuscript.

This manuscript has the feeling of many observations put together, maybe it's a summary of the work of a PhD or Masters thesis? But it's not in an adequate form for a paper. I wonder, all of this data must have been taken with a single particle, are the results repeatable? There are many nice results here, but the authors need to carefully rethink their story. I would be happy to look at this again.

General points

1. I'm not sure about your use of the term "rotational oscillations" early in the manuscript. By line 38 you are using the more precise "librational motion". If you don't want to be technical in the abstract, I would prefer "angular oscillations".
2. At the end of the paragraph starting on line 24, you make a bold statement about other researchers only managing to bring one degree of freedom to the ground state with the others uncontrolled. This isn't true... for example <https://doi.org/10.1038/s41567-023-01956-1> reports 2D ground state cooling, and <https://doi.org/10.1038/s41567-023-02006-6> reports 6D cooling (not to the ground state, but their angular degrees of freedom are cooled to the milliKelvin level).
3. You say that the damping on the librational frequencies (10Hz) is very low, but at the pressures which you are operating at they seem incredibly high! I would expect it to be below micro-Hertz, can you explain this to me? Similarly, you claim that the photon recoil noise on the librational motion will be lower than the translational, this sounds like a very complicated problem to me, do you have any literature or calculations to back this up?
4. Around line 71, you mention a "second mechanism" in the trapping. This sounds very interesting, but is not clearly explained, can you expand upon this point?

5. Regarding the precision of your size measurements, it took me a while to notice that you are extremely precise at measuring the ratio of different dimensions. What is your absolute accuracy? When comparing to the literature, I feel that maybe fair comparisons are not being drawn, e.g. your work looks very impressive compared to <https://doi.org/10.1063/5.0128606> (which should probably be cited), but they are reporting absolute measurements. I also believe that work such as <https://doi.org/10.1103/PhysRevA.102.013505> and <https://doi.org/10.1364/OPTICA.4.000356> provide high absolute dimension determination based on frequency analysis.

Your measurement also relies on an accurate measurement of pressure (to infer the mass of the particle), and you state in the Methods section that you use a capacitance gauge with an accuracy of 0.5%, I was under the impression that capacitance gauges were not reliable at very low pressures?

All in all, I am not convinced about the level of precision with which you state your determination of particle geometry, I think it is based on all kinds of measurements which cannot reach the stated level of precision, and also I'm not convinced that the *theory* is precise. The theory presented in the supplementary information even omits "higher order terms have no more than a few percent impact on the calculation results", but you're stating numbers at the 10ppm level.

6. If the depths of your librational potential is 100 Kelvin, why are they not spinning rapidly, as seen in work with other near-spheres? At your pressures, the rotation rates could be well above GHz, are you able to monitor at these frequencies? Your librational signals are clearly a couple of orders of magnitude smaller than your translational ones, which may mask the exact mechanism at play. For example, if your electric field actually *excites* the librational degrees of freedom, driving *occasional rotation* (which would lead to a smaller amplitude of the librational peak) this would have a complicated effect on the frequencies due to gyroscopic stabilization. Can you show a time-series of the librational motion (just for the review).

7. Further on the librational motion, I am surprised to see no mention of coupling between different degrees of freedom, which has limited other experiments attempting to cool librations. Especially because you are reporting nonlinear effects, where the different modes should couple! I would expect precession, nutation... the whole spectrum of interesting effects that appear when you have angular momentum involved.

8. The temperatures you achieve are good by the standards of feedback cooling (though, surpassed by cavity cooling <https://doi.org/10.1038/s41567-023-02006-6>), but is this not only because you are operating at a lower pressure? You're cooling is clearly limited by SNR, so there's no real indication of how good it could be.

9. Your method of thermometry seems highly specific to your exact experimental situation. You're saying that by monitoring the shift of the particle frequencies with respect to the PSD you can infer the temperature? However, doesn't the definition of temperature here rely on the system being a harmonic oscillator, but your system is an anharmonic oscillator? I'm also confused as to how your oscillations are so large that you sample the nonlinear part of the potential (as the frequencies are shifting), but the temperatures are low; is this just because the potential is so shallow? Can you estimate your mean angular deviation of motion?

10. I don't think you have enough evidence to support your claims in the paragraph starting on line 156. You have already hit your detection limit, reducing your pressure further will not help you

cool more. Or do you believe that the cooling is just happening without you “seeing” it? I don’t see what’s special about the near spherical particles here.

Small points

- Abstract line 7: I think the mentioning of “robotics” is irrelevant here, unless it is some kind of optically controlled nanorobot? Or something like this: <https://doi.org/10.1038/s41467-019-08968-7>
- Abstract line 10-11: “...where their quantum mechanical properties may be prominent”, subtle point, but there’s nothing particularly quantum about cooling to the ground state, maybe “from which their quantum properties may be made evident”.
- Abstract line 11: the reader may not know the significance of 6 degrees of freedom here, maybe you can use “all degrees of freedom” or if you wish “all external degrees of freedom”.
- In the paragraph starting on line 24, you make a comment about the “cost of having to tackle six mechanical degrees of freedom”. I disagree with this, levitated particles have a far simpler mode structure than any other oscillator I can think of, I believe people working with tethered oscillators can only dream of only six vibrational modes!
- In the paragraph starting on line 40, you make the point that nanospheres were only considered as candidates for translational control, but you show they can be considered for angular control... but that’s only because they aren’t actually spheres! I find this argument a bit circular.
- Line 103: moment of inertia.

Reply to referee 1

First of all, we would like to thank the referee for very careful reading and for making valuable comments that would improve our manuscript. The comments from the theoretical viewpoint is valuable for experimentalists. Below we list our replies to the comments. The line numbers corresponds to the revised manuscript (new sentences are shown in red characters). We welcome further comments if there are.

Referee's comment 1:

One of the main claims of the manuscript is that all three angles of a nearly spherical particle can be deeply trapped in a linearly polarized optical field if the tweezer is sufficiently anisotropic. While I can believe that the authors indeed observe trapping of all three rotational degrees of freedom, I am not convinced by this mechanism. I thus believe that this point requires further strengthening.

Reply

We added more detailed explanations on how the orientational confinement arises in lines 82-91. In addition, we added a new panel in Fig.1 to intuitively show the mechanism. In short, under the generalized Rayleigh-Gans approximation, we consider the inhomogeneous electric field of the trapping laser in the finite extent of the nanoparticle. The mechanical potential energy is obtained by integrating the potential energy density over the volume of the nanoparticle. The potential energy depends on the relative angle between the nanoparticle and the optical trap, and is minimized when the electric field experienced by the nanoparticle is maximized. This means that the long axis is aligned perpendicular to the optical lattice in our setup. The connection with previous theoretical studies is explained in detail in our reply to the comment 4.

Referee's comment 2:

Have the authors performed an independent check to verify that the three observed peaks are indeed due to rotations of the particle? Such a check could for instance use that the three angles are not independent [as Eq. (7) suggests; see below] but coupled non-linearly due to the gyroscope effect. Such gyroscopic coupling is one of the main signatures of rotations and is limiting other rotational cooling experiments thus far [PRL 127, 123605 (2021); PRR 2, 043054 (2020); Nat. Phys. <https://doi.org/10.1038/s41567-023-02006-6> (2023)].

Reply

Thank you for pointing out the important aspect. We indeed observe couplings between observed three peaks. We added sentences to explain the observed coupling behavior in lines 72-76 and 149-152. In addition, we show additional measurements on the correlation between each peak in terms of the oscillation frequency in supplementary information.

Referee's comment 3:

If the authors' interpretation is correct, the trapping frequencies should vary with the root of the tweezer power (as is also the case for the centre-of-mass frequencies). Has this been verified experimentally? Admittedly, this check would not prove the proposed mechanism, but it would at least rule out that the observed alignment is due to, e.g., an offset electrostatic field coupling to the permanent dipole moment of the particle.

Reply

Thank you for your careful consideration. We confirm the dependence on the laser intensity and show the results in the supplementary information. In fact, we observe that all the three

frequencies are proportional to the square-root of the laser power, similarly to the center-of-mass motions. This fact is now stated in the main text (lines 70-71).

Referee's comment 4:

The authors' interpretation that the particle aligns with the trap anisotropy is based on Eq. (SI_4) in the SI. What is the theoretical foundation of this equation? To derive it, the authors integrate the centre-of-mass trapping potential over the particle volume, arguing that this yields the 'trap potential energy at which the particle is tilted'. This statement makes no sense since in this equation x , y and z are the centre-of-mass coordinates of the particle, rather than field coordinates. If the authors' argument is that the electric field cannot be approximated as homogeneous over the extension of the particle (i.e., that the Rayleigh approximation fails), then this should be accounted for by a proper generalization of the optical torques derived for instance in [PRA 103, 043514 (2021)].

Reply

We are sorry that the derivation of our expressions are not very carefully shown. The theoretical foundation of our expressions are provided in previous theoretical studies, such as PRR 2, 033437 (2020) and PRA 94, 033818 (2016). In particular, [PRR 2, 033437] considers a standing-wave trap and is closely connected to our system. We confirm that our expressions agree with the analytical expression in [PRR 2, 033437] when the radius is sufficiently smaller than the wavelength and the beam waist. To be more precise, we derived higher order terms in the supplementary information and included them in our analysis in the revised manuscript, while the original expressions are left in Methods because they are useful for experimentalists (Eq.(3)). The agreement between theory and experiment in terms of librational frequencies is now even better than before.

Following the suggestion by the referee, we also examined the expressions of the torques shown in [PRA 103, 043514 (2021)], which is obtained by integrating Maxwell's stress tensor. As discussed in [PRR 2, 033437], the results obtained with the potential energy and Maxwell's stress tensor are expected to be the same for small particles, while we understand that the approach with Maxwell's tensor may be suitable for evaluating behaviors of larger particles with a size comparable to the wavelength (although numerically demanding). We think that with our radius (10% of the wavelength) the motions are still well described by analytical expressions based on the potential energy. We added sentences to explain these points in the main text in lines 82-91 and also in Methods (lines 291-299).

Referee's comment 5:

Throughout the manuscript, the authors treat the three angles as independent degrees of freedom. This is incorrect in general due to the gyroscope effect and only justified for deeply trapped rotations (librations). I am wondering why neglecting this coupling is justified even when non-linearities of the trapping potential become relevant. Specifically, if the equipartition theorem holds, as assumed in the analysis, this coupling should be on the same order of magnitude as the non-linearities of the trapping potential.

Reply

We indeed observe couplings between each peak. When we cool two of the three librational motions, the amplitude of the remaining peak is also reduced, but is not extinguished. The nonlinear frequency shift is only prominent at temperatures of higher than 1K, as shown in Figure 3 c,d,e. Therefore, we think that at low temperatures (<1K), librational motions are in the deep-trapping regime, if we borrow the word from [PRA 103, 043514], and are nearly independent from each other. We added these explanations in lines 72-76 and 149-152.

Regarding your comment on equipartition (we thank you for your insight), we do not explicitly assume it in our expression (5), while it is the case at the limit of A close to zero.

The kinetic energy and the potential energy are in general not the same. If you have further comments or suggestions on this point, we would appreciate them.

Referee's comment 6:

The discussion of the rotor dynamics in presence of trapping non-linearities is confusing because it does not distinguish clearly between fundamental parameters and time-averaged, effective quantities in the non-linear regime. For instance, I would propose to refer to Eq. (1) as the average frequency (or similar) rather than as the 'librational frequency' (which is already defined as ω_0).

Reply

Thank you for your suggestion. We modified as suggested (line 121 and line 304).

Referee's comment 7:

In the second equation in the SI I would expect a factor of 2 in front of the amplitude A in the numerator of the fraction because of the $\sin(2\psi)$ term on the right-hand side of the first equation in the SI.

Reply

We corrected. Thank you for pointing it out.

Referee's comment 8:

How is the replacement of the time-dependent sine function in the $SI = 1/\sqrt{2}$ justified? This seems very arbitrary, given that averaging the sine function over one oscillation period yields zero.

Reply

We are sorry for our confusing way of writing. If we expand the expression as the Taylor series, it can be written only via $\sin^2(2\pi f_{li} t)$. Therefore, we can replace it with $1/2$. We now explain it in the supplementary information.

Referee's comment 9:

The potential energy of a non-spherical particle in a laser field does not decompose into a sum as assumed in Eqs. (SI_1) – (SI_3), unless the angles can be harmonically expanded. For the correct expression (in the Rayleigh regime), see for instance [PRA 103, 043514 (2021)].

Reply

As explained in the reply to the comment 4, we carefully examined references, including the suggested one, and found that in fact our expressions are not exact, while they agree with previous theoretical results in e.g. [PRR 2, 033437 (2020)] when the radius is sufficiently smaller than the wavelength and the beam waist. We understand that the results on the suggested paper [PRA 103, 043514 (2021)] should also be the same for small particles and more correct for larger particles, while numerically evaluating the expressions is more demanding for us. We updated our arguments in the main text (lines 82-91) and in the Methods section (lines 291-299) and also provided complete derivations in Supplementary information.

Referee's comment 10:

The authors might want to consider referencing the works from Peter Barker's group on 6D cooling of nanoparticles [Nat. Phys. <https://doi.org/10.1038/s41567-023-02006-6> (2023)] and on measuring the shape of optically levitated nanorotors [APL 121, 221102 (2022)].

Reply

We added citations to the suggested references (lines 35-37, 43-44, 216-220, 242-244).

Reply to referee 2

First of all, we would like to thank the referee for very careful reading and for making various valuable comments that would improve our manuscript. Also, we appreciate the positive evaluation for the present work. Below we list our replies to the comments. The line numbers corresponds to the revised manuscript (new sentences are shown in red characters). We welcome further comments if there are.

Referee's comment 1:

Line 50: An error on the average radius of only 4 nm is derived, but as discussed in the supplementary material this number appears to assume a density of 2.2 g/cm³ with negligible error. The authors should comment on how this density is known with sufficiently small error, in particular since previous work has typically found lower density for the spherical nanoparticles used in most experiments, and exact measurement of that density is challenging. If there is an error on density, how would it translate to the predicted frequencies used to constrain the particle sphericity?

Reply

We greatly thank the referee for pointing out this important aspect. In fact, we just assumed a fixed density as has been the case with many other papers. We carefully re-examined our data and found that there are two independent measurements related to the radius and the density, but with different dependences on them: the heating rate due to background gas collisions at high pressures (about 5Pa), and the heating rate at high vacuum (about 10⁻⁶ Pa) determined by photon recoil heating. The latter is more sensitive to the radius. By combining these two measurements, we determine both the radius and the density. The density is now 2.26 g/cm³ with an error of 2%, which we understand is higher than found in larger particles and is close to the value of the bulk material. The influence of the error in density is now evaluated and provided in Table 1 (corresponding to about 0.01% in sphericity).

We added sentences to explain these aspects in Methods (lines 269-275). Furthermore, in relation with the reply to the comment 2, we mentioned the systematic uncertainties in the main text in lines 106-108. Also, we updated the value of the precision in determining the sphericity to 0.09% instead of 10 ppm.

Referee's comment 2:

Line 86 and elsewhere: The paper claims to measure the fractional ellipticity (q/p, r/p) at the 10 ppm level by comparing the observed frequencies and a model. However, I don't see any discussion of systematic errors relevant for this measurement. Instead the claim is it is primarily limited by the accuracy of fitting the frequency in the PSD. Could the authors comment whether e.g. small deviations in the polarization, shape of the optical potential, etc could provide relevant systematic errors for these fits at the ppm level? Have these systematics (or other dominant ones) been constrained and ruled out?

Reply

Again we thank the referee for pointing the important aspect. We carefully considered the sources of systematic uncertainties and listed on Table 1. Now we conclude that the dominant source of the systematic uncertainties are uncertainties in the refractive index and the measured radius. The value of precision is now modified to 0.09% in the sphericity. We added sentences to explain these considerations in the main text in lines 106-108 and in Methods (lines 319-321).

Referee's comment 3:

Line 134-138: The authors provide a fairly precise measurement of the dipole moment and its orientation. However, previous studies have found significant variations in the apparent dipole moment over time in larger optically trapped microparticles (e.g. Ref. 29). Since some conductivity is likely possible for charge within the particle, at least on long time scales, information quantifying the stability of such charge distributions may be useful if available. Such time variation may also be relevant to whether charges are primarily located on the surface as suggested in Lines 136-138.

Reply

Thank you for insightful comments. We have measured the stability of the dipole moment over half a day, and hardly observed its variation. If measured for a much longer period, as in PRA 106, 023503 (2022), we may observe some variations. We think this would be an interesting future direction, in particular to perform such measurements with a much smaller particle than in PRA 106, 023503 (2022). We added sentences to address this point in lines 170-171, 173-175.

Reply to referee 3

First of all, we would like to greatly thank the referee for very careful reading and for making various valuable comments that would improve our manuscript. The broad and deep insights of the referee help us greatly. Also, we appreciate positive evaluations such as “There are many nice results here...” as well as suggestions on optimal choices on wording. Below we list our replies to the comments. The line numbers corresponds to the revised manuscript (new sentences are shown in red characters).

We welcome further comments if there are.

Referee’s comment 1:

As a very general point, I think the authors need to consider what the point of this paper is. They do not achieve temperatures lower than others (for their librators), so this is not the selling point; it’s very far from the ground state of the librators. The cooling mechanism seems novel, but it’s barely explained! The size estimaton I think has major issues, similarly the thermometry is highly specfic and has issues. So, is the selling point that they are working with spheres, but they have achieved enough control and detecton efciciency to realise that there’s no such thing as a sphere? If so, why is this interestng (I’m not sure I buy the argument it’s “beter” for anything, but it’s certainly an observaton worth making)?

Reply

Regarding the selling point mentioned by the referee, we think that it is important that all the angular motions are now controlled for a particle with translational motions in the quantum regime (occupation number ~ 1). From this point of view, our work is a first demonstration. If one considers quantum mechanical experiments on the translational motions, the angular motions may mask interesting quantum effects, as stated in lines 224-230. We would like to refer to our latest arxiv submission relevant to this aspect (arXiv:2306.16598). The present work is therefore an important step for future quantum experiments. To clarify this point, we modified the abstract (line 14-16).

Regarding the question on why the present results are interesting, we would like to draw an attention that, up to now, only near-spherical particles have been brought to the quantum ground state of their translational motions. We therefore believe that establishing controls over their angular motions is very important in future studies. In fact, the present work demonstrates that the nearly spherical geometry possesses an advantage of simpler spectra than those of highly anisotropic particles, which will be beneficial for their controls in the quantum regime. To clarify these aspects, we added these explanations in lines 51-53 and modified the introduction in 31-33.

Referee’s comment 2:

The fgures in this manuscript are of a generally low quality. All fgures are too small, with crowded axes labels. Figure 1 is ok. Figure 2 is poor, very crowded, and barely shows the librational frequencies (just as an inset with many other unexplained peaks) even though this is the core of the paper. I feel that the fgures do not convey the core messages of the manuscript

Reply

We updated figures to meet the criticisms that they are not well-presented. We also added assignments on higher order signals in Figure 2.

Referee’s comment 3:

This manuscript has the feeling of many observatons put together, maybe it’s a summary of the

work of a PhD or Masters thesis? But it's not in an adequate form for a paper. I wonder, all of this data must have been taken with a single particle, are the results repeatable? There are many nice results here, but the authors need to carefully rethink their story. I would be happy to look at this .again

Reply

We presented the data that are required to convince readers. They are not just a summary of thesis. We believe that the revision significantly improved the manuscript. To meet the criticism that the presented results might be some rare events for a specific particle, we show additional measurements on the correlation among librational motions for another particle in supplementary information. Our observations are general to all the trapped nanoparticle in our system, and six-dimensional feedback cooling is realizable unless some of librational frequencies are coincidentally close to one of translational frequencies.

Referee's comment 4:

I'm not sure about your use of the term "rotational oscillations" early in the manuscript. By line 38 you are using the more precise "librational motion". If you don't want to be technical in the abstract, I would prefer "angular oscillations".

Reply

We modified the term as suggested.

Referee's comment 5:

At the end of the paragraph starting on line 24, you make a bold statement about other researchers only managing to bring one degree of freedom to the ground state with the others uncontrolled. This isn't true... for example <https://doi.org/10.1038/s41567-023-01956-1> reports 2D ground state cooling, and <https://doi.org/10.1038/s41567-023-02006-6> reports 6D cooling (not to the ground state, but their angular degrees of freedom are cooled to the milliKelvin level).

Reply

Thank you for the important point out. We missed the paper on the 6D cooling on arXiv (which was really our big mistake), while both the paper on 6D cooling and the paper on the 2D cooling were officially published after the submission of the present work. We added citations to both literatures (lines 35-37, 43-44, 242-244). In addition, we modified the sentence to meet the comment (lines 35-37).

Referee's comment 6:

You say that the damping on the librational frequencies (10Hz) is very low, but at the pressures which you are operating at they seem incredibly high! I would expect it to be below microHertz, can you explain this to me? Similarly, you claim that the photon recoil noise on the librational motion will be lower than the translational, this sounds like a very complicated problem to me, do you have any literature or calculations to back this up?

Reply

We would like to point out that the spectral width of 10Hz is a value obtained on the power spectral density and only reflects the Fourier limit. The spectral width is also readily broadened via fluctuations in the laser intensity and in the amplitude of librational motions. To clarify these points and to avoid misleading, we modified the sentence in lines 182-183. The calculations on photon recoil torques for various geometries are given in e.g. PRA 95, 053421 (2017).

Referee's comment 7:

Around line 71, you mention a "second mechanism" in the trapping. This sounds very interesting, but is not clearly explained, can you expand upon this point?

Reply

Thank you for your interest. We added detailed explanations on the mechanism in lines 82-91 as well as a panel in Fig.1 to show an intuitive picture.

Referee's comment 8:

Regarding the precision of your size measurements, it took me a while to notice that you are extremely precise at measuring the ratio of different dimensions. What is your absolute accuracy? When comparing to the literature, I feel that maybe fair comparisons are not being drawn, e.g. your work looks very impressive compared to <https://doi.org/10.1063/5.0128606> (which should probably be cited), but they are reporting absolute measurements. I also believe that work such as <https://doi.org/10.1103/PhysRevA.102.013505> and <https://doi.org/10.1364/OPTICA.4.000356> provide high absolute dimension determination based on frequency analysis.

Your measurement also relies on an accurate measurement of pressure (to infer the mass of the particle), and you state in the Methods section that you use a capacitance gauge with an accuracy of 0.5%, I was under the impression that capacitance gauges were not reliable at very low pressures?

All in all, I am not convinced about the level of precision with which you state your determination of particle geometry, I think it is based on all kinds of measurements which cannot reach the stated level of precision, and also I'm not convinced that the theory is precise. The theory presented in the supplementary information even omits "higher order terms have no more than a few percent impact on the calculation results", but you're stating numbers at the 10ppm level.

Reply

Thank you for pointing out the very important aspect. We now added explanations on systematic uncertainties in the Methods section (lines 319-321, table 1). The theory is now updated to include higher order terms such that it does not limit the determination of the geometry. The agreement between theory and experiment in terms of librational frequencies is now even better. We concluded that the accuracy in determining the ratio of the radii is about 0.09 %, which is dominantly limited by the uncertainties in the refractive index and in measured mean radius. The value of the precision in determining the shape given in the main text is also updated. We added these explanations in lines 107-108.

From the referee's comment, we became aware that showing the precision in length as "corresponding to" is confusing and therefore such a statement was deleted, while comparisons with other studies in length are left in lines 218-220.

Regarding your comment on the capacitance gauge, we use it to measure the radius at around 5Pa where the gauge is the most sensitive. We also thank the point out on relevant literatures, which are cited now.

Referee's comment 9:

If the depths of your librational potential is 100 Kelvin, why are they not spinning rapidly, as seen in work with other near-spheres? At your pressures, the rotation rates could be well above GHz, are you able to monitor at these frequencies? Your librational signals are clearly a couple of orders of magnitude smaller than your translational ones, which may mask the exact mechanism at play. For example, if your electric field actually excites the librational degrees of freedom, driving occasional rotation (which would lead to a smaller amplitude of the librational peak) this would have a complicated effect on the frequencies due to gyroscopic stabilization. Can you show a time-series of the librational motion (just for the review)

Reply

Thank you for your elaborate consideration. We would like to point out that spinning occurs when the light is circularly polarized and provides angular momentum to the nanoparticle. Because we use a linearly polarized light, we expect that spinning would not occur. In fact, we have never observed signals at a frequency range above a few 100 kHz apart from the center-of-mass oscillation along the optical lattice. Feedback controls on librational motions via an applied electric field show a clear dependence on its phase, from which we can clearly identify if it is heating or cooling the librational motion. To more clarify this point, we added sentences in lines 145-146. We also draw an attention that the time variation of the amplitude of librational motion under feedback cooling is in good agreement with theory (Fig.4a). The agreement also strengthens our argument, which is also mentioned in line 165-166. We are willing to provide the data on the time-series of the librational motion as follows (but now for another particle).

Figure: **Time series of the librational motion** Time series of the signal of the librational motion around the y axis. The signal is obtained by passing the photodetector signal through a high-pass filter at 15 kHz and a low-pass filter at 53 kHz.

Referee’s comment 10:

Further on the librational motion, I am surprised to see no mention of coupling between different degrees of freedom, which has limited other experiments attempting to cool librators. Especially because you are reporting nonlinear effects, where the different modes should couple! I would expect precession, nutation... the whole spectrum of interesting effects that appear when you have angular momentum involved.

Reply

In fact, we observe couplings between librational motions, which are prominent at large amplitudes. We added sentences to explain the observed behaviors in lines 72-76 and 149-152.

Referee’s comment 11:

The temperatures you achieve are good by the standards of feedback cooling (though, surpassed by cavity cooling <https://doi.org/10.1038/s41567-023-02006-6>), but is this not only because you are operating at a lower pressure? Your cooling is clearly limited by SNR, so there’s no real indicator of how good it could be.

Reply

The low temperature achieved is partially thanks to the low pressure, but also related to the fact that our feedback cooling is not parametric feedback cooling (driven at twice the oscillation frequency), but is cold damping (driven at the oscillation frequency). From the theory of cold damping (e.g. PRL 122, 223601 (2019)), the ultimate limit is determined by the

compromise between the heating rate and the signal-to-noise ratio (increasing the feedback gain decreases the effect of intrinsic heating, but also results in heating due to the noise in feedback itself), implying that the temperature can be made arbitrarily low until it reaches the ground state, if signal-to-noise ratio is infinitely high. In reality, of course, the signal-to-noise ratio is finite and limits the achievable temperature. To clarify these aspects, we added sentences in lines 144-145, 154-157.

Referee's comment 12:

Your method of thermometry seems highly specific to your exact experimental situation. You're saying that by monitoring the shift of the particle frequencies with respect to the PSD you can infer the temperature? However, doesn't the definition of temperature here rely on the system being a harmonic oscillator, but your system is an anharmonic oscillator? I'm also confused as to how your oscillations are so large that you sample the nonlinear part of the potential (as the frequencies are shifting), but the temperatures are low; is this just because the potential is so shallow? Can you estimate your mean angular deviation of motion?

Reply

We are sorry for confusing the referee. What we mean is as follows: the extent of the nonlinearity is a direct measure of the absolute value of the angular deviation (if we borrow your word), when the amplitude variation is very slow as is the case with our system. The model on our thermometry relies on the fact that the system is an anharmonic oscillator, and we define the temperature of librational motions without assuming a harmonic oscillator (as given in supplementary information).

We became aware that a similar idea has recently been implemented with the center-of-mass motions of levitated objects. Therefore, the method itself is a quite general approach without relying on thermalization. We now added a sentence to clarify the issue and explain recent studies with levitated particles in line 131-133.

Regarding your question on sampling the nonlinearity, we would like to draw an attention that the frequency shift is clearly visible only at relatively high temperatures. As suggested by the referee, the low potential depth is important to clearly observe the shift. When feedback cooling is turned on and the temperature is lower than 1K, we hardly observe the nonlinear frequency shift, as shown in Figure 3 c,d,e.

To clarify these aspects, we added sentences in lines 134-135.

The mean angular deviation is about 4 degrees at a temperature of 1K for the librational frequency at 14 kHz (corresponding to $f_{\{lx\}}$).

Referee's comment 13:

I don't think you have enough evidence to support your claims in the paragraph starting on line 156. You have already hit your detection limit, reducing your pressure further will not help you cool more. Or do you believe that the cooling is just happening without you "seeing" it? I don't see what's special about the near spherical particles here.

Reply

Thank you for your honest consideration. In many quantum experiments, low decoherence is a crucial ingredient to sustain quantum states. Our claim is that near-spherical particles have significantly small photon recoil heating, an intrinsic decoherence source with optical trapping. Theoretically, the ground-state cooling of librational motions is not required to observe quantum behaviors (e.g. New J. Phys. 20, 122001 (2018)). In addition, we do not think that our setup is already at the best condition for observing librational motions; rather, it is just that we somehow observe additional motions in a setup dedicated to translational motions. If we consider the history of cooling the translational motions, it was initially limited to of the order of 10 mK, and later improvements in experimental setup significantly reduced

the temperatures. We anticipate that a similar improvement will enable us to further enhance the signal-to-noise ratio for observation and to reduce the temperature.

We moved the sentence for discussing future possibilities to the concluding remarks in line 234 and 236-239, while we leave discussions on slow heating of librational motions in line 195-201.

Referee's comment 14:

Abstract line 7: I think the mentioning of "robotcs" is irrelevant here, unless it is some kind of optcally controlled nanorobot? Or something like this: <https://doi.org/10.1038/s41467-019-08968-7>

Reply

We corrected as suggested.

Referee's comment 15:

Abstract line 10-11: "...where their quantum mechanical propertes may be prominent", subtle point, but there's nothing particularly quantum about cooling to the ground state, maybe "from which their quantum propertes may be made evident"

Reply

We corrected as suggested. Thank you for suggesting a suitable way of writing.

Referee's comment 16:

Abstract line 11: the reader may not know the signifcance of 6 degrees of freedom here, maybe you can use "all degrees of freedom" or if you wish "all external degrees of freedom".

Reply

We corrected as suggested.

Referee's comment 17:

In the paragraph staring on line 24, you make a comment about the "cost of having to tackle six mechanical degrees of freedom". I disagree with this, levitated partcles have a far simpler mode structure than any other oscillator I can think of, I believe people working with tethered oscillators can only dream of only six vibratonal modes

Reply

We agree with the referee and deleted the sentence.

Referee's comment 18:

In the paragraph startng on line 40, you make the point that nanospheres were only considered as candidates for translational control, but you show they can be considered for angular control... but that's only because they aren't actually spheres! I fnd this argument a bit circular.

Reply

We agree and corrected as follows: "near-spherical nanoparticles, whose deviation from a sphere has been overlooked in previous studies on manipulating the translational motions at the quantum level".

Referee's comment 19:

Line 103: moment of inertia

Reply

We corrected the term.

List of modifications from the initially submitted version

Here we list all the modifications made in the revised version. The line numbers are based on the revised manuscript. In the revised manuscript, all the modified words/sentences are shown in red characters.

1. In the abstract, wording is modified according to the suggestions. In addition, we modified the main selling point in line 14-16.
2. The upper limit of the temperatures of librational motions is modified from 0.1K to 0.03K (line 20,153,205) to be more precise.
3. The value of precision in determining the ratio of radii is modified from 10ppm to 0.09% based on considerations on various systematic uncertainties.
4. According to suggestions, the sentence in line 28-29 was deleted.
5. In line 31-33, we modified a sentence to be more precise (nearly-spherical particles instead of nanospheres) and to clarify the selling point.
6. We mentioned very recent related studies (published after the submission of the present work) in line 35-37, 43-44, and 242-244.
7. In line 45-46, the sentence is modified to meet the criticism.
8. In line 51-53, we added a sentence to emphasize the advantage of our system to meet the criticism that the selling point is unclear.
9. The error in determining the radius is updated. The reasoning is carefully explained in Methods (line 269-275).
10. In line 70-71, we added our observation on the frequency variation with respect to the laser intensity to clarify the raised question.
11. In line 72-76 and 149-152, we added explanations on the correlation among librational motions to clarify the raised question.
12. In line 82-91, we added careful explanations on how orientational confinement occurs in our setup to clarify the raised question.
13. In line 104-105, we added an explanation to clarify the measurement condition.
14. In line 106, 354, 371, 372, 387, 388, and in the caption of Figure 3, the values of the ratio of radii are updated because of the inclusion of higher order terms to clarify the raised question. In addition, because of the inclusion of higher order terms in the theory, the agreement between theory and experiment in terms of frequency in line 107 is also updated.
15. In line 107-108, we modified the value of precision and added the origin of systematic uncertainties to clarify the raised question. Accordingly, wording in line 109 is modified.
16. In line 120-121 and in line 324-325, we added a sentence for the assumption in the model. The wording in line 121, in 304, and in line 327 was also modified.
17. In line 96, we corrected the sentence on f_{li0} . Accordingly, the definition of f_{li0} in line 123-125 was deleted. Similarly, we corrected the expressions on f_{li0} in Eq.(3,6) and in line 309.
18. In line 131-133, we added an explanation on how our thermometry works and cited recent related studies.
19. In line 134-135, we added a sentence to clarify the raised question on nonlinearity.
20. In line 144-145, 154-157, we added sentences to clarify why our cooling is efficient. In addition, in line 145-146, we added an explanation that cooling is dependent on the phase to strengthen our argument.
21. In line 161 and 166-167, the words „due to feedback cooling“ is added to be more precise.
22. In line 165-166, we added a sentence to strengthen our argument to clarify the raised question.

23. In line 170-171 and 173-175, we added sentences to address the stability of the dipole moment to clarify the raised question.
24. In line 182-183, we added an explanation on how the spectra width is broadened to clarify the raised question.
25. In line 195-196, we added a sentence to strengthen our argument to meet the criticism, while arguments on future possibilities are moved to line 236-239.
26. In line 216-217, 218-220, we modified sentences to compare our results with previous studies.
27. In line 221-222, we modified wording to make it more understandable.
28. In line 236-239, we added a sentence to meet the criticism.
29. In line 269-275 in Methods, we modified sentences to explain how we determine the radius and the density of nanoparticles used in the present work.
30. In line 291-299 in Methods, we added explanations on theoretical foundations of the orientational confinements.
31. In line 319-321 in Methods, we added a statement of the systematic uncertainties in the ratio of radii.
32. In Figure 1, we added a new panel c that shows an intuitive picture of orientational confinement in an anisotropic trap. Accordingly, the caption is added.
33. Figure 2-4 are updated to meet the criticism that they are difficult to see. In particular, the inset in Figure 2 is now shown as a new panel, where we show assignments on higher order signals.
34. In Figure 3, the measurement condition is added on the caption.
35. In Extended Data section, table 1 is added to show various mechanisms for systematic uncertainties.
36. In the acknowledgement section, we added a funding that was missed in the initial manuscript.
37. Supplementary information is updated to support all the arguments.

REVIEWER COMMENTS

Reviewer #1 (Remarks to the Author):

I thank the authors for their detailed reply. I find all their arguments convincing and have not further scientific questions/concerns to the manuscript content. I thus recommend publication of the manuscript as it is.

Reviewer #2 (Remarks to the Author):

I would like to thank the authors for their consideration of my previous comments and detailed response. Overall, I believe the manuscript has been improved compared to the previous version. However, I am still concerned about the accuracy of the reported results and errors.

The authors have addressed my primary comment that the errors on various parameters used in their model fits were not specified, and in particular have included Table 1. The new errors reported on the particle shapes (the accuracy of which is a central result of the paper, highlighted in the abstract) are now reported to be substantially larger than in the initial draft. However, my assessment is that this updated accounting is not sufficient to justify the results at the reported accuracy, at least without more details.

In particular, the authors now identify that errors in the knowledge of the refractive index may be the dominant systematic error on the radius, but which were not accounted for in the initial draft. This dominant error is then addressed in only a single sentence in line 320: "...refractive index, which is assumed to be 1%,...". This statement is given without justification or further details. Unless the authors have details not in the manuscript, it is very unclear to me how they know the refractive index to the percent level. Very few details are given about the particles themselves. Most experiments using silica nanoparticles rely on particles grown through sol-gel processes, which are known to differ in their optical properties from optical grade fused silica. As the leading systematic, I believe that an arbitrary assumption of 1% error is not sufficient to gain confidence on the propagate errors to other parameters in the analysis. I would recommend the authors clarify this issue (and possibly any other errors that could significantly affect their reported results) before publication.

Reviewer #3 (Remarks to the Author):

Document uploaded

I thank the authors for the hard work they put into improving this manuscript. I note that all reviewers had significant overlapping comments, and also that the authors have worked hard to address all points.

I am happy to recommend publication, though I believe the work is of interest only to the levitated optomechanics community, and it's up to the Editor whether that is acceptable for Nature Communications.

1. Regarding my point regarding the focus of the work, I believe the authors have sensibly settled on the concept that nearly everyone works with near-spheres, and so this work will be relevant to most of the community.
2. Regarding the figures, Figure 2 is now much improved. The other figures are still small and crowded, but this is probably not enough to preclude publication.
3. I apologise for my comment regarding the nature of the data presentation, as I feel this may have come across as rude. There is repeated data from a second particle in the supplementary information. This is minimally sufficient.
4. Point addressed.
5. Point addressed.
6. Point addressed.
7. Point addressed.
8. I still have my reservations about this, especially since the librational frequency is broadened by noise effects, but if it is accepted by the other referees I will accept it.
9. I accept this argument. For your information, I may expect the particle to spin even in linearly polarized light because Brownian motion / thermal fluctuations could drive rotations of the particle when the librational potential is very shallow (less than room temperature). If your COM potential depth was 100K, then you would lose the particle due to thermal fluctuations, even though there is a restoring force towards the centre.
10. Point addressed.
11. Point addressed.
12. Point addressed.
13. Point addressed.
14. All further points addressed.

Reply to referee 2

First of all, we would like to thank the referee again for very careful reading and for making various valuable comments that would improve our manuscript. Below we list our reply to the comments.

Referee's comment 1:

The authors have addressed my primary comment that the errors on various parameters used in their model fits were not specified, and in particular have included Table 1. The new errors reported on the particle shapes (the accuracy of which is a central result of the paper, highlighted in the abstract) are now reported to be substantially larger than in the initial draft. However, my assessment is that this updated accounting is not sufficient to justify the results at the reported accuracy, at least without more details.

In particular, the authors now identify that errors in the knowledge of the refractive index may be the dominant systematic error on the radius, but which were not accounted for in the initial draft. This dominant error is then addressed in only a single sentence in line 320: "...refractive index, which is assumed to be 1%,...". This statement is given without justification or further details. Unless the authors have details not in the manuscript, it is very unclear to me how they know the refractive index to the percent level. Very few details are given about the particles themselves. Most experiments using silica nanoparticles rely on particles grown through sol-gel processes, which are known to differ in their optical properties from optical grade fused silica. As the leading systematic, I believe that an arbitrary assumption of 1% error is not sufficient to gain confidence on the propagate errors to other parameters in the analysis. I would recommend the authors clarify this issue (and possibly any other errors that could significantly affect their reported results) before publication.

Reply

We thank the referee for pointing out an important aspect that was not very carefully addressed in the previous manuscript. We admit that the error in the refractive index presented in the previous manuscript was not convincing.

To make the arguments more convincing, we now present the data of measurements on the refractive index of the sample used in the present work (details are given in supplementary information). In short, we measure the transmissivity of the mixture of the SiO₂ sample and the liquids with precisely measured refractive indices. We observe a high transmissivity only when the refractive index of nanoparticles agrees with the liquid.

Using the measured refractive index, we updated all the calculations. The values of ratios of radii are also updated. The density value, which was updated in the last revision, was also slightly modified. In terms of librational frequencies, the agreement between calculations and experiments is even better than in the previous manuscript.

From the width of the measured peak, we derive a new estimation on the systematic error in the refractive index, which is now slightly smaller than before (0.76%). Accordingly, the error in the ratios of radii is also updated.

These modifications are explained in line 316-320 in the revised manuscript.

We also checked other systematic errors and confirmed that they are reasonable.

Reply to referee 3

First of all, we would like to thank the referee again for very careful reading and for making various valuable comments that would improve our manuscript. We are also grateful to accepting all the points. Although there are no major comments, we provide our replies to some of the comments.

Referee's comment 2:

Regarding the figures, Figure 2 is now much improved. The other figures are still small and crowded, but this is probably not enough to preclude publication.

Reply

We modified the figures such that they are easier to see. We hope that they are fine now.

Referee's comment 8:

I still have my reservations about this, especially since the librational frequency is broadened by noise effects, but if it is accepted by the other referees I will accept it

Reply

We would like to draw attention that the systematic error in determining the librational frequency, listed in Table 1, is not the dominant source of the error in determining the ratios of radii.

The largest uncertainties are the errors in the refractive index and in the radius. Referee 2 provided a new comment on this point. While the error in radius measurements is properly given, the error in the refractive index was not appropriate in the previous manuscript (we used the refractive index of the bulk SiO₂ material, without knowing to what extent the refractive index of nanoparticles is deviated).

Therefore, we performed measurements of the refractive index of the particle sample used in the present work. Using the measured refractive index, we updated all the calculations. In terms of librational frequencies, the agreement between calculations and experiments is even better than in the previous manuscript. The systematic error in the refractive index is also updated. Accordingly, we updated the value of the systematic error in determining the ratios of radii.

Referee's comment 9:

I accept this argument. For your information, I may expect the particle to spin even in linearly polarized light because Brownian motion / thermal fluctuations could drive rotations of the particle when the librational potential is very shallow (less than room temperature). If your COM potential depth was 100K, then you would lose the particle due to thermal fluctuations, even though there is a restoring force towards the centre.

Reply

We thank the referee for the information. From our side, we agree with the expectation of the referee, while we also think that rotational motions, instead of librational motions, rarely occur, because the phase for feedback cooling the librational motion is stable for many hours. We rarely observe the flip of the phase for feedback cooling, which may indicate the situation suggested by the referee.

List of modifications from the initially submitted version

Here we list all the modifications made in the revised version. The line numbers are based on the revised manuscript. In the revised manuscript, all the modified words/sentences are shown in red characters.

1. In line 105-106, the values of the ratios of two radii and the deviation in librational frequencies between calculations and experiments are updated by using the measured refractive index value. Similarly, the values of the ratios are updated in the caption of Figure 3, Extended Data Figure 2, and Extended Data Figure 5, and in line 353. Based on new calculations, we also updated the plots in Fig.3(a), Extended Data Fig.2, 4, 5.
2. In line 316-320, we added explanations on the independent measurements of the refractive index of the SiO₂ sample used in the present work. The value of the refractive index as well as its systematic error are updated. The value of the radius was also updated, while its error stays the same as before. Accordingly, the systematic error given in Table 1 and the value of the systematic error in determining the ratios in line 18, 107, and 215 is also updated.
3. The sizes and layouts of all the figures are modified to meet the comment by referee 3 (comment 2).
4. The grant name and number was recently updated.

REVIEWERS' COMMENTS

Reviewer #2 (Remarks to the Author):

I would like to thank the authors for their response to the previous comments, and for including details regarding a direct measurement of the refractive index of the particles used. I have no further concerns and would recommend publication of the revised manuscript.

Reviewer #3 (Remarks to the Author):

I am happy with the scientific analysis in this report now.